# "Good Robot! Now Watch This!": Repurposing Reinforcement Learning for Task-to-Task Transfer

**Andrew Hundt**[1]* **Aditya Murali**[1]* **Priyanka Hubli**[1] **Ran Liu**[1]

**Nakul Gopalan**[2] **Matthew Gombolay**[2] **Gregory D. Hager**[1]
[1]Johns Hopkins University  [2] Georgia Institute of Technology
* Equal contribution {ahundt,amurali6}@jhu.edu

**Abstract:** Modern Reinforcement Learning (RL) algorithms are not sample-efficient to train on multi-step tasks in complex domains, impeding their wider deployment in the real world. We address this problem by leveraging the insight that RL models trained to complete one set of tasks can be re-purposed to complete related tasks when given just a handful of demonstrations. Based upon this insight, we propose See-SPOT-Run (SSR), a new computational approach to robot learning that enables a robot to complete a variety of real robot tasks in novel problem domains *without* task-specific training. SSR uses pretrained RL models to create vectors that represent model, task, and action relevance in demonstration and test scenes. SSR then compares these vectors via our Cycle Consistency Distance (CCD) metric to determine the next action to take. SSR completes 58% more task steps and 20% more trials than a baseline few-shot learning method that requires task-specific training. SSR also achieves a *four order of magnitude improvement in compute efficiency* and a *20% to three order of magnitude improvement in sample efficiency* compared to the baseline and to training RL models from scratch. To our knowledge, we are the first to address multi-step tasks from demonstration on a real robot without task-specific training, where both the visual input and action space output are high dimensional. Code is available in the supplement.

## 1 Introduction

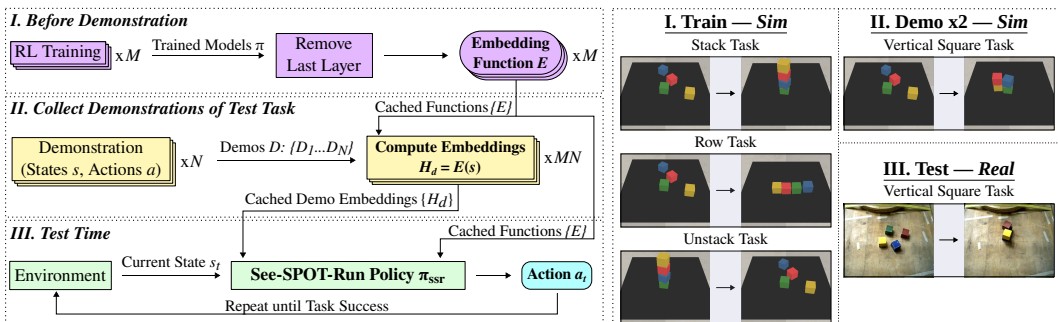

Figure 1: **Left:** Reinforcement Learning Before Demonstration (RLBD) paradigm. **Right:** One of four folds for leave-one-out cross-validation with sim to real transfer and a novel test task.

The ability to quickly adapt deep Reinforcement Learning (RL) models to new tasks with just a handful of demonstrations would dramatically enhance their practicality and applicability. Repurposing previously-trained models for new tasks consumes several orders of magnitude fewer computational resources than training from scratch. For our approach to repurposing RL models, we draw inspiration from social imitation and learning in animals. Certain animals, dogs, for example, have a strong capacity to imitate human performance of a novel task when trained with the "Do as I Do" method [1, 2]. Imagine a dog named Spot that can be been trained with positive reinforcement

5th Conference on Robot Learning (CoRL 2021), London, UK.

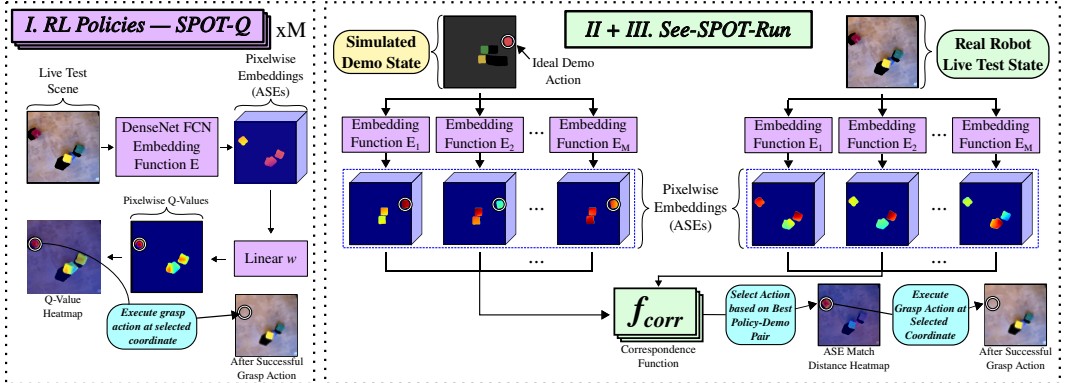

Figure 2: **Left:** An outline of our pretrained RL policies trained with SPOT-Q [4]. Each RL policy contains an Embedding Function $E$ that outputs pixel-wise (per action) embeddings (16x224x224x64 or $H^{\Theta \times X \times Y}$ ASEs). These embeddings are turned into Q-values (16x224x224x1) by a linear layer. The action space $A$ has the same dimension as the Q-values (16x224x224x1, or $\Theta \times X \times Y$) for each `grasp` and `place` action. **Right:** See-SPOT-Run (SSR) Framework architecture executing a row task, the grasp action is selected from the vertical square policy. SSR successfully executes novel tasks using a correspondence function $f_{corr}$ (Eqs. 4-8) that matches the demonstrated action to the live test scene using the embeddings (ASEs) derived from each pre-trained policy.

("Good Spot!") to imitate a new action shown to her, such as fetching a specific object from a container, when prompted by a command ("Watch This!"). When Spot imitates, "it is the orientation and the shape of the action [that is] attended to and reproduced" [2, 3]. Spot encodes what she sees in an internal representation, which she may then map onto her trained skillset in order to successfully imitate a new task.

We observe that some deep RL models, like Spot the dog, *already* encode meaningful representations of a scene that can be mapped to a skillset. We hypothesize that we can leverage these encodings to solve new tasks from demonstration *without task-specific learning*, a concept we call **Reinforcement Learning *Before* Demonstration** (RLBD, Fig. 1). To evaluate our hypothesis, we propose the See-SPOT-Run Framework for RLBD.

See-SPOT-Run (SSR, Fig. 2) rapidly adapts an ensemble of existing RL policies to new tasks by imitating demonstrations. Intuitively, existing fully-trained and validated RL models are already invariant to certain changes in the scene within the tested circumstances. We pass a known state and a novel state to a preexisting model to create and then compare latent vectors in a search for embedded task-invariant knowledge to use in novel circumstances. When the known state is from a novel task demonstration, and the novel state is from a live test scene, we can pinpoint latent vectors that map to actions in the test environment that are likely to progress towards completing the novel demonstrated task. Therefore, we apply See-SPOT-Run to imitate demonstrations of a new multi-step task ("Watch This!") by matching demonstrated actions to candidate actions in a test environment. In doing so, we not only enable few-shot imitation of unseen tasks but also provide the potential to vastly expand robotic skill sets at dramatically lower costs.

In our few-shot imitation experiments, we demonstrate See-SPOT-Run's ability to complete a set of multi-step tasks, including block stacking, row making, block unstacking, and building a vertical square structure (see Fig. 1). These tasks are an order of magnitude more challenging than typical reach or limited grasp and place tasks [4] (see Sec 3). To evaluate each task, we use RL policies trained to complete different, and even conflicting (stacking vs unstacking) tasks, and provide two expert demonstrations collected in simulation. In summary, our contributions are as follows:

1. Reinforcement Learning Before Demonstration (RLBD): We propose a concept to use pretrained RL policies to accelerate robotic imitation learning. We show that these policies generate Action-State Embeddings (ASEs) from robot state observations, and that these embeddings can be re-purposed to execute unseen tasks.

2. See-SPOT-Run (SSR): Our RLBD implementation that accomplishes both few-shot imitation and zero-shot sim-to-real domain transfer by matching successful demonstrated actions to the test environment with a novel *Cycle Consistency Distance (CCD)* metric.

3. SSR achieves competitive task performance without the task-specific training required by prior work, leading to a *four order of magnitude improvement in compute efficiency* and a *20% to three order of magnitude improvement in sample efficiency* for novel tasks when compared to the baseline and to training RL models from scratch.

## 2  Related Work

Deep learning approaches have successfully completed a range of robotic tasks in visually-guided manipulation [5, 6] and navigation with high sample complexity [7, 8, 9, 10, 11, 12, 13]. VPG [6] achieved particularly efficient RL training for decluttering tasks. "Good Robot!" [4] improved on VPG's RL sample efficiency and showed zero-shot sim-to-real transfer for multi-step block tasks with a risk of *progress reversal* [4], *e.g.* toppling a partially completed tower (see Sec. 3); however, it requires from-scratch task-specific training.

Another approach to tackling deep RL's high sample complexity is through multi-task and meta-learning [14, 15, 16, 17, 18, 19, 20] of general purpose task representations or policies that allow an agent to quickly adapt to novel tasks by fine-tuning previously learned policies. James et al. [16] approach the few-shot learning problem by first meta-learning a task embedding network from a set of training tasks, then computing a test task embedding from a few demonstrations; this task embedding is provided as input to a control policy. Meanwhile, Finn et al. [14] perform one-shot imitation on tasks seen during meta-training but evaluated on unseen objects. While these approaches achieve good performance on one-step tasks, they do not consider multi-step tasks or *progress reversal*. Alternately, few-shot inverse RL methods [21, 22, 23] aim to increase RL sample efficiency by learning a reward function from one or a few demonstrations, then using this reward function to supervise RL training. Yet, Liu et al. [21] and Singh et al. [22] do not address learning multi-step tasks, which can require dense reward functions to learn effectively [4, 10]. Goo and Niekum [23], in turn, propose an IRL approach for multi-step tasks that requires time and compute-intensive training. Zhu et al. [24] reviews Transfer Learning, and our work can be framed as Representation Transfer.

Cycle Consistency is a mechanism for comparing representations across domains [25, 26, 27]. Zhang et al. [28] utilizes Cycle Consistency for cross domain dynamics in simulated locomotion tasks and sim-to-real transfer of a basic robot reaching task, showing that Cycle Consistency can mitigate domain shift for the same, single-step, task. Our See-SPOT-Run framework, by contrast, completes both sim-to-real *and* known-to-novel task transfer with more complex multi-step tasks via our own cycle consistency concept.

Behavior Cloning (BC), Imitation Learning (IL) [29, 30, 31, 32, 33, 34, 35], and IL from Observation (ILO) [31], are approaches to mimic the behaviors of agents. Recent reviews cover IL [36, 37] and ILO [38]. Jung and Kim [39] take a hybrid BC and IL approach to grasp and place one block on another block. GAIL [40] learns to imitate with inspiration from generative adversarial networks. Others pursue distribution matching approaches [33, 25, 30]. However, these BC, IL, and ILO papers variously require training on novel tasks, are evaluated only on simulated tasks, or do not consider multi-step tasks with a risk of *progress reversal* [4]. NTP [41] and NTG [42] learn task plans in a blocks environment by imitating videos, but require a perception pipeline to act.

TransporterNets [29] models multi-step IL tasks characterized by *progress reversal* as a sequence of displacements via behavior cloning, with impressive results that scale from 1-1k demonstrations per task. However, TransporterNets makes no attempt at task-to-task transfer for different tasks. Instead, it requires real-world, task-specific demonstrations and training for every unique task, so their approach is simultaneously much easier to solve (direct task transfer) and much more difficult to adapt to new scenarios (due to task-specific training). In comparison, SSR addresses the problem of few-shot imitation with no task-specific training. Even with these limitations, TransporterNets is the closest available method to SSR and consequently, we chose it as our baseline for comparison.

## 3  Preliminaries

As in "Good Robot!" [4] and VPG [6], we investigate multi-step tasks by framing our problem as a Markov Decision Process (MDP) $(S, A, P, R)$, with state space $S$; action space $A$; transition probability function $P : S \times S \times A \to \mathbb{R}$; and reward function $R : S \times A \to \mathbb{R}$. We make an MDP-simplifying assumption where states equal observations, and that there is a discrete action space. At time step $t$, the agent observes state $s_t$ and chooses an action $a_t$ according to its policy $\pi : S \to A$, leading to a new state $s_{t+1}$ with probability $P(s_{t+1}|s_t, a_t)$. Q-Learning is an RL method whose

purpose is to seek a $Q$ function that maximizes $R$ over time. Q-Learning policies $\pi$ select an action $a_t$ at time $t$ by first estimating the reward $R$ with a function $Q : S \times A \to R$, then selecting the action that maximizes the expected reward: $a_t = \pi(s_t) = \operatorname{argmax}_{a \in A} Q(s_t, a)$.

Our See-SPOT-Run Framework experiments are pretrained with the Q-Learning-based SPOT-Q framework introduced by "Good Robot!" [4]. SPOT-Q demonstrates the importance of prioritizing *progress reversal* during RL training, and also effectively integrates 'common-sense' exploration constraints into Q-learning. Our RL policies are pretrained with the same exploration constraints and reward functions $\mathbb{R}_{\text{trial}}$, $R_{\mathcal{P}}$ as the best performing models in [4]. In our development, we make use of a task progress measure, $\mathcal{P} : S \to \mathbb{Z}^{[0,p_{\max}]}$ as an integer indicating the number of successful task steps; the task is complete when $\mathcal{P}(s_t) = p_{\max}$. Consequently, a *progress reversal* [4] occurs when an action $a_t$ at a timestep $t$ undoes previous task progress *i.e.* there exists an $i \in [0, t]$ such that $\mathcal{P}(s_{t-i}) > \mathcal{P}(s_{t+1})$. Next, we will introduce our proposed methods.

## 4   Methods

SSR has three phases: **(1) Embed**, **(2) Correspond**, and **(3) Act**.

In phase **(1) Embed**, SSR takes a set of fully-trained policies, such as Q-functions, and converts them into *embedding functions* by extracting the latent per-action feature vectors embedded in the final network layer immediately before Q-values are created (Fig. 2). SSR runs these *embedding functions* on every demonstration and test state to make a unique latent space for each (`policy`, `demonstration`) pair.

Phase **(2) Correspond** has two steps: (2a) Match and (2b) Verify. In (2a), SSR compares the embedded demonstration state to the embedded test state of each latent space, searching for the test action that most closely matches the successful demonstration action. It finds this closest match by minimizing the euclidean distance between the demonstration action's embedding (feature vector) and each potential test action's embedding. Since matches are made in the embedding space, the matched action should mirror the qualities of the demonstrated action for the new task, and need not bear any resemblance to the action the original preexist-

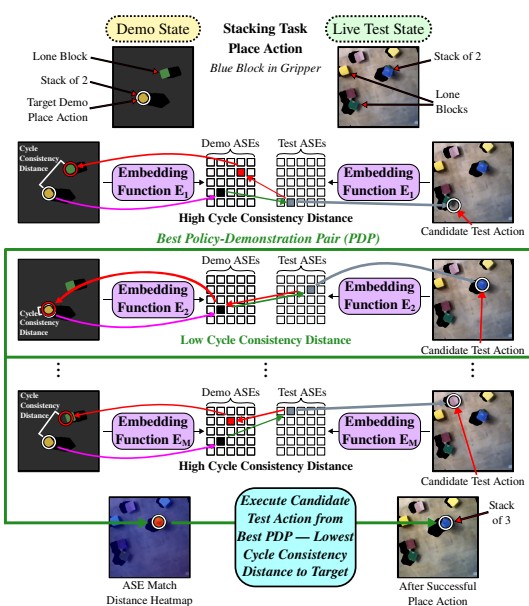

Figure 3: An illustration of cycle consistency correspondence for 3 policy-demonstration pairs (each row is a pair) as See-SPOT-Run (SSR, Fig. 2, Eq. 8) chooses to place a hidden blue block already in the gripper onto the blue block visible in the test state image.

ing policy would choose for its original task. Intuitively, the high dimensionality of latent vectors ensures dissimilar actions do not collide, in a manner analogous to hash functions. For example, we successfully choose reasonable action matches within policies and across states (Sec. 5), but matches across policies were random and unsuccessful in small tests. Ultimately, this matching process yields a set of candidate actions, one for each (`policy`, `demonstration`) pair.

In (2b) Verify, SSR reverses the process, starting with each test candidate action, retrieving its embedding, and performing the match process in the demo scene to see how that action would perform in the demo world. The physical cartesian distance between the known ideal demo action and our new rematch demo action in the action space is our *cycle consistency distance* metric (Fig. 3); intuitively, actions that minimize this distance are more likely to accomplish the demonstrated behavior.

Finally, in phase **(3) Act**, SSR selects the candidate action that minimizes the cycle consistency distance and acts. It repeats this entire process until task completion. SSR thus conducts a comprehensive search over all (`policy`, `demonstration`) pairs, jointly selecting the latent space and demonstration that are most relevant to the novel task as well as the action that will most likely advance task progress. In Section 4.1, we will describe how we construct the embedding functions, then in Section 4.2, we will elaborate on how SSR uses these embedding functions to imitate demonstrations, and provide additional intuition for our cycle consistency distance metric.

## 4.1 Generating Embedding Functions

We begin by pretraining task-specific SPOT-Q [4] policies $\pi_m : m = 1...M$ on $M$ known tasks: Recall that Q functions are defined $v_a = Q(s, a)$; we reframe this as $V = Q_m(s)$, where $V$ contains scores for every action $a \in A$ pertaining to task $m$. Thus, the Q-value of taking action $a$ in state $s$ is $Q_m(s)[a]$. Finally, from each $Q_m$, we strip off the last linear layer $w_m$, a linear projection (dense layer) that maps a feature vector to a scalar Q-value, to obtain as many **Embedding Functions**[1] $E_m$:

$$\pi_m(s) = \underset{a \in A}{\operatorname{argmax}} Q_m(s, a) = \underset{a \in A}{\operatorname{argmax}} Q_m(s)[a] = \underset{a \in A}{\operatorname{argmax}} E_m(s)[a] \cdot w_m \tag{1}$$

Evaluating each $E_m$ on an arbitrary state yields a latent space array $H_m$, composed of one **Action-State Embedding (ASE)** $h_{m,a} = H_m[a]$ for each action space coordinate $a \in A$, as in Fig. 2.

## 4.2 Test-Time See-SPOT-Run Policy

In this section, we will define our test-time SSR policy, $\pi_t^{\text{ssr}}$, which observes the test state $s_t$ and selects an action $a_t$ for the robot to execute, as it attempts to complete the demonstrated test task. It solves the *imitation problem*: approximating an unknown optimal live test task action $a_t^*$ that will complete the next step of a novel, demonstrated task. SSR uses information from $N$ demonstrations $D : \{D_1, ..., D_N\}$. Each demo $D_n : \{(s_{\text{demo}}, a_{\text{demo}})_1, ..., (s_{\text{demo}}, a_{\text{demo}})_T\}$ is a sequence of State-Action pairs which maximize $R$ at each task step $t \in T$.

**(1) Embed:** We begin by observing the test state $s_t$ and progress $p = \mathcal{P}(s_t)$ and we consider the $N$ state-action pairs $(s_{\text{demo},n}, a_{\text{demo},n})$ with progress $p$ from the N demonstrations[2]. We use these to compute latent space representations of each demonstration state and the test state with each embedding function $E_m$, yielding $M \times N$ demo latent space arrays $H_{m,n}$, and $M$ test latent space arrays[3] $H_{m,t}$:

$$H_{m,n} = E_m(s_{\text{demo},n}) \qquad (2) \qquad\qquad H_{m,t} = E_m(s_t) \qquad (3)$$

**(2) Correspond:** We find candidate test actions (2a), then verify these candidate actions with our cycle consistency distance metric (2b). To find a candidate action, we match the demo action to the test state by selecting the minimum euclidean distance between the demo action embedding (ASE) and all ASE vectors in the test latent space. We repeat this comparison for each latent space, yielding $M \times N$ total candidate actions $F_{m,n}$, one for each (`policy`, `demonstration`) pair. Candidate actions are expressed as coordinates in the action space $A$, and can be used to index quantities such as the latent spaces $H$, which have the same dimension as $A$, as detailed in the Fig. 2 caption.

$$L_{m,n} = \min_{a \in A} \| H_{m,t} - H_{m,n}[a_{\text{demo},n}] \|_2 \quad (4) \quad F_{m,n} = \underset{a \in A}{\operatorname{argmin}} \| H_{m,t} - H_{m,n}[a_{\text{demo},n}] \|_2 \quad (5)$$

We then verify each matched candidate test action in the demo scene using our *cycle consistency distance* metric. We verify, or match in reverse, the candidate action's ASE $H_{m,t}[F_{m,n}]$ to the corresponding demonstration latent space $H_{m,n}$, minimizing euclidean distance in the latent space as above. We expect the verification action $a_{\text{rematch}}$ to be physically close to the original demonstrated action $a_{\text{demo},n}$ in the action space $A$. Therefore, we define the cycle consistency distance $C_{m,n}$ as the Cartesian distance between $a_{\text{demo},n}$ and $a_{\text{rematch}}$.

$$a_{\text{rematch}} = \underset{a \in A}{\operatorname{argmin}} \| H_{m,t}[F_{m,n}] - H_{m,n}[a] \|_2 \tag{6}$$

$$C_{m,n} = \| a_{\text{demo}} - a_{\text{rematch}} \|_2 \tag{7}$$

The purpose of the cycle consistency distance $C_{m,n}$ is to consider other ASE vectors in the latent space $H_{m,n}$ that might indicate that a candidate action $F_{m,n}$ is not an authentic match. Alternatively, the rematch step can be interpreted as an assessment of how relevant a given latent space is to the test state and task at hand.

---

[1]In equations, parenthesis ( ) are for function arguments and brackets [ ] index arrays or discrete coordinates.

[2]We note that, because each target demonstration action $a_{\text{demo}}$ is optimal, demos satisfy the property that they have monotonically increasing progress. Our method is trivially extended to suboptimal demonstrations by considering all state-action pairs with progress $p$.

[3]While Section 4.1 refers to latent spaces as $H_m$, we extend this notation here, using $H_{m,n}$ to designate the state from the $n$-th demonstration embedded with $E_m$. Similarly, $H_{m,t}$ is the test state embedded with $E_m$.

**(3) Act:** We select and execute the candidate action with minimal Cycle Consistency Distance (Eq. 8). We also evaluate a baseline SSR without cycle consistency by skipping the rematch phase and directly minimizing the *L2 Consistency Distance* (Eq. 9) $L_{m,n}$.

$$\hat{m}, \hat{n} = \operatorname*{argmin}_{m,n} C_{m,n} \qquad\qquad \hat{m}, \hat{n} = \operatorname*{argmin}_{m,n} L_{m,n}$$

$$a_t^{\mathrm{ssr}} = F_{\hat{m},\hat{n}} \quad (8) \qquad\qquad a_t^{\mathrm{ssr}} = F_{\hat{m},\hat{n}} \quad (9)$$

This completes our definition of the SSR policy $a_t^{\mathrm{ssr}} = \pi_{\mathrm{ssr}}(s_t)$. Every time an action $a_t^{\mathrm{ssr}}$ completes we collect a new observation $s_{t+1}$, running the SSR policy $\pi_{\mathrm{ssr}}$ repeatedly until task progress $\mathcal{P}(s_T)$ reaches its maximum, $p_{\max}$, indicating that the task is complete. We evaluate See-SPOT-Run with L2 consistency distance (**SSR L2CD** in Tables 1, 2, and 3) and cycle consistency distance (**SSR CCD** in Tables 1, 2, and 3), as well as prior work, with few-shot experiments in Sec. 5. We will also discuss how our SSR Framework with CCD surpasses the other methods.

## 5 Experiments

We outline our assessment metrics in Sec. 5.1, provide simulation results in Sec. 5.2, then cover our sim-to-real transfer results in Sec. 5.3. We pretrain models with SPOT-Q [4] (Sec. 3) with the workspace, commands and action space defined in "Good Robot!" [4]. Implementation and Robot details are in Fig. 1 and Appendix C.1.

### 5.1 Evaluation Metrics

Our metrics quantify the broad improvements in task performance and reductions in the resources necessary to perform novel tasks. Most important, we consider critical efficiency measures that have not previously been evaluated in the baselines, to motivate the broad range of useful applications for both RBLD and SSR.

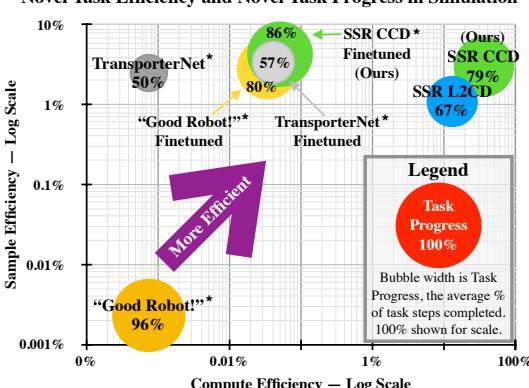

Figure 4: Higher is better on all metrics (Sec 5.1). Log Scale. Larger bubbles are better. See Tab. 1.

Efficiency Metric Performance

| | Data References | Trial Success |
|---|---|---|
| **SSR L2CD** | 3.0000% | 13% |
| Sample Efficiency | 1.1000% | |
| Average Task Progress | 67.0000% | |
| **SSR CCD** | 36.0000% | 36% |
| Sample Efficiency | 3.0000% | |
| Average Task Progress | 79.0000% | |
| **TransporterNet** | 0.0008% | 30% |
| Sample Efficiency | 2.5000% | |
| Average Task Progress | 50% | |
| **"Good Robot!"** | 0.0008% | 91% |
| Sample Efficiency | 0.0028% | |
| Average Task Progress | 96% | |
| **TransporterNet Fine-tuned** | 0.0410% | |
| Sample Efficiency | 3.4000% | |
| Average Task Progress | 57% | |
| **SSR CCD Finetuned** | 0.051% | |
| Sample Efficiency | 4.3000% | |
| Average Task Progress | 86% | |

**Test Metrics** evaluate how effectively the robot completes the test tasks, and higher is better: (1) **Trial Success Rate (Trials)** is the percentage of multi-step tasks completed 100% successfully, and in many applications completing a task is a prerequisite to moving on to the next task. (2) **Action Efficiency (Eff.)** is the $\frac{\text{ideal}}{\text{actual}}$ number of actions, more efficient models will complete tasks with fewer actions. [6] Our ideal is 6 actions for all tasks except for rows, which is 4 actions. [4] (3) **Progress (Prog.)** is each trial's maximum proportional within-task progress, averaged over all trials, *e.g.* a 3 of 4 block stack is 75%, to show capabilities that nearly complete trials. (4) **Recoveries (Recov.)** is the percentage of trials *in which there was a mistake such as a progress reversal* that the agent was able to complete with a trial success, *i.e.* $\frac{\text{successes containing } progress\ reversal}{\text{trials containing } progress\ reversal}$. Higher recovery rates reflect better robustness to perturbations and uncommon situations.

**Cost Metrics** delineate resources spent, and lower is better: (5) **Train Steps** is the number of neural network batch steps performed prior to executing on a novel test task. Each individual experiment is run on one NVIDIA GeForce RTX 2080Ti GPU. (6) **Annotated Actions (Ann. Actions)** is the number of robot actions $a_t$ that have been annotated by a human or scripted observer, which are each fairly high cost activities in robot or human time [43, 44].

**Efficiency Metrics** evaluate trial success benefits with respect to the cost metrics, and higher is better. (7) **Compute Efficiency** $\frac{\text{Trials}}{\text{Train Steps}+1}$ is the percentage of trials that can be completed for every training batch step when completing a novel task. We add one to the train steps denominator to prevent dividing by 0 with SSR. (8) **Sample Efficiency** $\frac{\text{Trials}}{\text{Ann. Actions}}$ measures the increase in trial success rate amortized over annotated actions on a test task.

### 5.2 Simulation Experiments

We pretrain RL policies on four tasks as in Fig. 1: stacking, row-making, unstacking, and 2x2 vertical square. At test time we perform four-fold cross-validation where the test task is given two

demonstrations but no pretrained model, then the remaining three models predict actions for the untrained test task. For the finetuned case, we pretrain each of the three models on two demonstrations of the untrained test task for 333 steps each, for a total of 1k steps. We then apply SSR as described in Sec. 4. We additionally investigate a "Good Robot" finetuning baseline where we use one of the finetuned models directly as the test-time policy, without applying SSR.

To obtain our TransporterNet [29] baselines on their less challenging scenarios (Sec. 2), we collect two demos of each test task in their *ravens* framework and train TransporterNet on these demos for 40k iterations; meanwhile, for the pretrained case we fine-tune their vertical block triangle model [29] on the novel task for 1k iterations (thus matching the overall training cost of our pretrained method). We will examine overall performance on the test metrics, task-specific test metrics of particular interest, and then our efficiency metrics.

| Simulation Task | Average Test Metrics | | | | Costs | | Efficiency | |
|---|---|---|---|---|---|---|---|---|
| | Trials | Action Efficiency | Prog. | Recov. | Train Steps | Annotated Actions | Compute | Sample |
| **SSR CCD Eq. 8 (ours)** | **36%** | **41%** | **79±1%** | **19%** | **0** | 12 | **36%** | **3.0%** |
| SSR L2CD Eq. 9 | 13% | 30% | 67±1% | 5% | 0 | 12 | 13% | 1.1% |
| TransporterNet [29]* | 30% | 35% | 50±3% | 8% | 40k | 12 | 0.00075% | 2.5% |
| **SSR CCD Eq. 8 Finetuned (ours)*** | **51%** | 35% | **86±1%** | **30%** | 1k | 12 | **0.051%** | **4.3%** |
| TransporterNet [29] Finetuned* | 41% | 36% | 57±3% | 12% | 1k | 12 | 0.041% | 3.4% |
| "Good Robot" [4] Finetuned* | 34% | **43%** | 80±1% | 16% | 1k | 12 | 0.034% | 2.83% |
| "Good Robot!" [4] * | 91%* | 57%* | 96±1%* | 90%* | 120k | 40k | 0.00076% | 0.0023% |

Table 1: Simulation task performance on the metrics detailed in Sec. 5.1, averaged over all four folds of leave-one-model-out cross-validation. "Average Test Metrics" averages Table 2 values. Bold indicates the best performing model. Higher is better for all metrics except costs. The progress range, *e.g.* in 50±3%, the 3 is standard error. * Starred methods address the simpler problems described in Sec. 2, so comparisons should carefully consider this context. TransporterNets [29] trains on robot demos for each novel task with no task-to-task transfer. SPOT-Q [4], the SSR pretraining step, tests on the train task, provides a cost and efficiency baseline plus a test metrics ceiling; finetuned SPOT-Q is 1k steps of tuning from a random task to the novel task on two demonstrations.

| Simulation Task | Stack | | | | Unstack | | | | Row | | | | Vertical Square | | | |
|---|---|---|---|---|---|---|---|---|---|---|---|---|---|---|---|---|
| | Trials | Action Eff. | Prog. | Recov. | Trials | Action Eff. | Prog. | Recov. | Trials | Action Eff. | Prog. | Recov. | Trials | Action Eff. | Prog. | Recov. |
| **SSR CCD Eq. 8 (ours)** | 24% | 22% | 75% | 16% | 66% | 77% | 82% | – | **30%** | 38% | 81% | 28% | 24% | 28% | 79% | 14% |
| SSR L2CD Eq. 9 | 0% | – | 51% | 0% | 38% | 60% | 76% | – | 6% | 27% | 73% | 7% | 8% | **34%** | 67% | 7% |
| TransporterNet [29]* | 12% | **27%** | 51% | 12% | **86%** | 63% | **94%** | – | 2% | 25% | 24% | 2% | 20% | 29% | 32% | 9% |
| **SSR CCD Eq. 8 Finetuned (ours)*** | 58% | 29% | **90%** | **42%** | **72%** | 64% | **87%** | - | **40%** | 19% | 84% | **24%** | **34%** | **28%** | **83%** | **26%** |
| TransporterNet [29] Finetuned* | 54% | **34%** | 74% | 12% | 56% | 34% | 64% | – | 24% | **26%** | 44% | 16% | 28% | 28% | 46% | 8% |
| "Good Robot" [4] Finetuned* | **64%** | 33% | 80% | 38% | 59% | **86%** | 82% | - | 20% | 24% | **90%** | 11% | 0% | 28% | 68% | 0% |

Table 2: Leave-one-model-out cross-validation of our See-SPOT-Run (SSR) framework for 50 simulation trials. Bold indicates key best metrics (Sec. 5.1). Unstacking has no notion of recovery.

Our results on the overall **Test Metrics** in Table 1 show SSR with CCD achieving 36% trial completion, 79% average progress, 41% action efficiency, and 29% recovery rate; L2 Correspondence L2CD gets 13%, 67%, 30%, and 5%, respectively, which demonstrates the benefit of evaluating demo embeddings $H_d$ with Cycle Consistency Correspondence CCD (Alg. 2); and TransporterNet gets 30%, 50%, 35%, and 8%, respectively, which highlights our method's overall improvement over prior few-shot imitation work.

Our task-specific simulation results in Table 2 show SSR with CCD performs better than L2CD in all cases, and better than TransporterNets in all cases except unstacking. The SSR worst case trial success rate of 24% is 12x better than the TransporterNets worst case of 2%. SSR completes 66% of unstacking trials vs 86% for TransporterNets, and SSR completes 30% of rows vs 2% for TransporterNets. This highlights a shortcoming of TransporterNets' feature template displacement matching approach which is not present in SSR. A more detailed per-task simulation breakdown is in Appendix C.3.

Lastly, the finetuning experiments show that SSR with CCD is effective not only in a zero-training setting, but also as an extension to few-shot model finetuning. Compared to the baseline finetuned "Good Robot", our SSR CCD Finetuned achieves 86% vs 80% average progress, 51% vs 35% trial completion, 35% vs 43% action efficiency, and 30% vs 16% recoveries. This highlights the mipact of the SSR Test Time policy (Sec. B.3), as SSR with Finetuning and "Good Robot" finetuning use the same base models and finetuning method. Moreover, SSR with Finetuning also outperforms

TransporterNets with Pre-Training, achieving 86% vs 57% average progress, 51% vs 41% trial completion, 35% vs 36% action efficiency, and 30% vs 12% recoveries.

**Efficiency Metrics**: With CCD, **SSR achieves 36% compute efficiency, which is roughly four orders of magnitude better** than TransporterNet and "Good Robot!", which both have 8e-4% compute efficiency. This is triple the 13% compute efficiency of L2CD, further underlining the effectiveness of our CCD metric. **SSR also achieves three orders of magnitude better sample efficiency** with CCD than 'Good Robot!", at 3% vs. 2e-3%, while maintaining a much smaller, but still significant, lead over TransporterNet at 2.5% sample efficiency, and L2CD at 1.1% sample efficiency. The smaller gap is because, as an imitation method, TransporterNets is able to replicate behaviors with very just a few annotated actions. Even so, SSR maintains its lead over TransporterNets due to its superior trial completion rate. Next we will examine our performance on the real robot experiments.

### 5.3 Real Robot Experiments

We transfer SSR to a real robot using our models pretrained in simulation from Section 5.2 with results in Tab. 3. All other aspects of the method remain the same except task progress is recorded by a different observer, since in simulation we read internal simulator states. TransporterNets is not designed for sim to real transfer, and is thus not included here.

In real experiments SSR has an average of 40% of trials complete, 42% action efficiency, 84% progress, 35% rate of recoveries, 40% compute; which is very similar to our results in simula-

| Real Task | Trials | Action Efficiency | Prog. | Recov. |
|---|---|---|---|---|
| Stack | **30%** | 25% | **80±5%** | 22% |
| Unstack | **90%** | 86% | **97±3%** | – |
| Row | **30%** | 28% | **77±7%** | 66% |
| Square | **10%** | 27% | **75±4%** | 10% |
| **Average** | **40%** | 42% | **82±3%** | 33% |

Table 3: Real See-SPOT-Run framework with Cycle Consistency, SSR CCD Eq. 8, performance on Sim-To-Real transfer to novel tasks from simulated demos during leave-one-model-out cross-validation. Bold entries are for ease of reading and progress range ($\pm$) is standard error.

tion at 36%, 41%, 79%, 29%, and 36% respectively. For reference, "Good Robot" completed 100% of trials after 20k RL training actions and 60k training steps for both stacks and rows, resulting in 0.0008% compute and 0.003% sample efficiency; but it scored on the train task, and is neither designed for nor capable of completing new test tasks on its own. We examine per-task real SSR test metrics in Appendix C.4.

## 6 Conclusion

See-SPOT-Run's three to four order of magnitude sample efficiency and compute efficiency improvements show it can *immediately* and *effectively* begin making progress on a novel task in a novel domain with scarce demonstration examples. By contrast, RL models such as "Good Robot!" typically start out with highly randomized exploration and no ability to complete a novel task, requiring inefficient sampling and compute-intensive training to first match and then beat SSR on a given task. Furthermore, although SSR performs 12x better than TransporterNets [29] in the worst case sim trials, TransporterNets may scale better than SSR on some metrics as demonstrations are added. Even so, it would ultimately be more efficient in the long run to develop methods where demonstrations on one multi-step task also improve performance and efficiency on the next task.

We expect that our proposed methods will be generally applicable to enhancing other methods across broad contexts such as Reinforcement Learning [4, 6], Behavior Cloning, Imitation Learning, and Meta-Learning; as well as in specific contexts such as TransporterNets. Future work should investigate such applications. Further improvements to our methods might include better correspondence metrics, automatically assessing task progress, mitigating the need for a dense reward [23, 26], and an extension to continuous domains.

In summary, we have introduced the concept of Reinforcement Learning Before Demonstration, implemented via the See-SPOT-Run framework. Our framework achieves simultaneous few-shot imitation and sim-to-real transfer for multi-step tasks, such as block stacking and row-making, that are prone to *progress reversal*. See-SPOT-Run improves *compute and sample efficiency by three to four orders of magnitude* vs. prior work by leveraging pretrained RL models to solve the *imitation problem* on novel tasks.

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
