# OpenReview forum: ""Good Robot! Now Watch This!": Repurposing Reinforcement Learning for Task-to-Task Transfer"
_robot-learning.org/CoRL/2021/Conference — CoRL2021 Poster_

### Official Review · Reviewer_YJsz · 2021-07-07

**Originality:** Good
**Technical Quality:** Very Good
**Clarity Of Presentation:** Fair
**Impact:** 4

**Recommendation:**

Weak Accept: I recommend accepting the paper, but will not argue for my recommendation if the majority of other reviewers have a different opinion.

**Summary:**

This paper proposes the Reinforcement Learning Before Demonstration (RLBD) paradigm for transfering agent policies to novel tasks given a few demonstrations but without task-specific training. The key idea is that 1) learning some embeding functions E by RL training; 2) compute the embeddings of demonstrations H_d = E(d) and testing state  H_l = E(s); 3) find the most ''relevant'' H_d for H_l and take the action corresponding to H_d under the current testing state. The authors propose two methods (i.e., L2 Consistency Distance and Cycle Consistency Distance) to measure the relevance of H_d and H_l. The evaluation results are much better than baselines.

**Issues:**

1) What if the best action in demonstration cannot cover the testing scenarios? could your methods deal with this?


2) How to train the embedding function E? Because there are totally M number of embedding functions, how to make them diverse enough? Or diversity is necessary? Training a large number of  embedding functions is also computational extensive.

**Reviewer Expertise:**

Fair: Some knowledge of the area

**Strengths And Weaknesses:**

Strengths:
1) a reasonable transfer framework to handle novel tasks.
2) good evaluation results.


Weaknesses:
1) poor writing. The method can only be understood with very careful reading, not because the method itself is complex, but due to the writing is poor. For example,  there are many short notes like 'SSR, Fig.2, Alg. 1, PDP, CTA ...'.

**Summary Of Recommendation:**

The idea is good and the results are encouraging, especially for the zero-shot sim-to-real domain transfer and real robot testing. Thus, I prefer to accept this paper.

However, my major concerns are that 1) the writting is poor and 2) the experiments are relatively simple.

Overall, I give a weak accept for this paper.

---

### Official Review · Reviewer_Hszk · 2021-07-23

**Originality:** Good
**Technical Quality:** Good
**Clarity Of Presentation:** Good
**Impact:** 2

**Recommendation:**

Strong Accept: I recommend accepting the paper and will argue for my recommendation even if other reviewers hold a different opinion.

**Summary:**

This work proposes the SSR (See-SPOT-Run) algorithm. Given a set of k pre-trained policies, SSR can generalize to a new robotic manipulation task given a small number of demonstrations and no fine-tuning. SSR does this by measuring the cycle-consistency distance between the demonstrations and new task in multiple learned embedding spaces. SSR is evaluated on block arrangement and stacking tasks in simulation and on a physical robot.


**Issues:**

Please address the points raised above on generality, the comparison with TransporterNets and robustness, as well as the minor issues.

**Reviewer Expertise:**

Good: General knowledge of the area

**Strengths And Weaknesses:**

**Strengths**

* Tackles an important problem in robot learning - how to generalize to new tasks with minimal new data and training requirements.
* SSR is interesting approach that is an original application of cycle consistency techniques. To my knowledge this is the first time it has been applied to task generalization in robotics, although I am not very familiar with the cycle consistency literature.
* Strong results demonstrating zero shot transfer to new tasks
     1. These are demonstrated both in simulation and on a real robot system, with comparable performance between simulation and the real world.
     2. The transfer from sim-to-real is also impressive.
* Related work is thorough.
* The cycle consistency approach is potentially quite general, although I have some questions given the current presentation (discussed below).

**Weaknesses**

Generality
* SSR is applied only to the SPOT framework which is a particular reward shaping methodology aimed at incorporating progress reversal into policy rewards. Consequently I question generality of the method.
* It would be great to see SSR applied to policies that are pre-trained differently (for example using TransporterNets or a Q-Learning method). Demonstrating that SSR yields strong performance across multiple methods for training the task-specific policies would be valuable.
* Similarly, it would be great to see how SSR performs for tasks not characterized by policy reversal.
    1. The method as proposed does not appear restricted to policy reversal tasks. The paper would be stronger if evidence was provided for the effectiveness of SSR in a robotics domain with different characteristics. For example, navigation which typically involves long episodic time horizons and continuous control. This would offer a good contrast short time horizon (~4 - 6 steps (optimally)) discrete action space problems considered.
* Reliance on a predefined task progress metric to identify the relevant demonstration action to use also limits the generality of SSR
    1. A discussion of to what extent this is a limitation and how this might be overcome would be a valuable addition to the paper.

Limited assessment of SSR compared with TransporterNets
* Given that TransporterNets are the only other method that SSR is compared with, it would have been nice to see a more extensive comparison of the two methods. Especially since the average success rates of the two methods are quite similar, at 30% and 36%.
* In particular, an ablation study which explores the effect of the number of demonstrations / annotated actions on both TransporterNet and SSR’s performance would be very valuable.
    1. As-is with the results presented using just 2 demonstrations, I question whether the lower performance of TransporterNets is simply due to the very low data regime.
    2. The paper would be stronger if SSR demonstrated consistently better performance across a range of data regimes, for example from 1 - 100 or 1000 annotated actions. The authors do mention this as future work, but I think it is important to include in this paper, especially since a large enough dataset appears to be already available.

The lack of pre-training for TransporterNets limits the fairness of the comparison.
* The SSR policies shown in Table 1 have seen 40k actions (albeit from different tasks), whereas the TransporterNets policies have only seen 12. Given the importance of data for training any neural network, it is possible that pretraining the TransporterNets could significantly improve performance. Please consider adding the following results or discussing why they are not relevant.
    1. TransporterNets, zero -shot: A baseline in which TransporterNets were pre-trained in the same way as SSR with 40k actions for 120k steps.
    2. TransporterNets, fine-tuned: A baseline in which TransporterNets were pre-trained in the same way as SSR with 40k actions for 120k steps, then fine-tuned on the 12 task-specific new actions.

Robustness
* No discussion is given on policy variance for a particular task or across training runs.
    1. At a minimum stating the number of trials per task that each policy was evaluated on and including the standard deviation of performance over these trials is important to include.

Minor:
* It would be nice to see how a Q-Learning method (e.g. Rainbow) performs on this task without the SPOT reward shaping. Similarly for an on-policy method. However I recognize time and compute constraints so do not view this as critical.
* The method is a little hard to understand, especially in sections 4.2 and 4.3
    1. Section 4.2: The notation is confusing, especially the re-framing of the Q function given that the Q function has a formal definition. Additionally the use of V is confusing given the Value function in RL: V^pi(s_t).
    2. Section 4.2:  E_m (s_t) → H_m. Isn’t the embedding also a function of the action?
    3. Section 4.3: Line 173: What does (4) refer to? Link between all the phases listed is lines 159 - 161 and the text in the section is hard to follow.
* Inaccuracies in Q function description
     1. Line 108: The Q function does not maximize R (reward) over time. It is defined as the expected cumulative discounted future rewards from state s_t given action a_t under policy pi
    2. General: The Q function is defined with respect to a particular policy, i.e. Q^pi
* Typos:
    1. Line 107: “seeks” → “learns”
    2. Line 267: “Transporternets” → “TransporterNets”
* Line 217: Which prior work?
* Line 233: Please reference that the Efficiency Metric was defined in [4]
* Line 281: Please give a little more detail as to which robot you are using.



**Summary Of Recommendation:**

Thank you for your submission. This work proposes a promising approach to the important problem in robotic learning of task generalization. The application of cycle-consistency to zero-shot task generalization is especially interesting. The zero-shot results are promising in simulation and in the real world, as is the sim-to-real transfer. The paper would be even stronger if the issues regarding method generality and the comparison with TransporterNets were addressed.

---

### Official Review · Reviewer_FEvJ · 2021-07-23

**Originality:** Very Good
**Technical Quality:** Excellent
**Clarity Of Presentation:** Fair
**Impact:** 4

**Recommendation:**

Strong Accept: I recommend accepting the paper and will argue for my recommendation even if other reviewers hold a different opinion.

**Summary:**

The paper proposes a few-shot learning methods for rapidly adapting to a new task using an ensemble of policies (and demonstrations) from old tasks. The essential idea is to consider already-acquired policies (Q-functions) to itself be themselves embedding functions which generate a score function E(s) -> H(a). To perform on a novel task given a couple demonstrations, we can embed the demonstration states using the policies for already-acquired tasks. We embed the test-time state in the same manner, and then choose new actions to most closely match (in the latent space) successful actions in from demonstration (which are presumed to be optimal). This search is exhaustive over `(policy, demonstration)` pairs.

In the variation presented here, the Q-functions are specifically learned to generate an a pixel-wise action-value function, to make computing the distance metric easier and more appropriate for the state space.

If policies (Q-functions) encode task semantics which are invariant to the irrelevant state of the environment, this should allow us to make forward progress on new tasks, by always choosing the "best" action which we already know works well on other tasks in a new task.




**Issues:**

So why the Weak Accept? Because I believe this paper has major presentation issues which will severely abridge its potential impact. If the authors revise the writing the paper for far greater intuition, easier reading, and conceptional clarity, this paper is clearly a strong accept.

Firstly, it relies heavily on a prior work for conceptual clarity -- it would be difficult to understand this paper without reading [1,2].

Second, within the body of this paper itself, the exposition of the motivation is compelling, but the authors fail to develop much intuition for their design choices, either referring to previous papers, or simply stating the design choices without motivating them with intuition. This makes it difficult for most readers to understand a paper with algorithmic designs which are much more complicated than a simple training loop.

Third, the exposition of the method itself is dripping with novel acronyms and custom-named concepts, only a fraction of which seems necessary for understanding the core ideas of the paper. These make it extremely difficult to move through the paper as a ready looking to answer her own questions, because each answer requires referring back in the text several times to understand the meaning and context of each named concept.

Fourth and more minorly, the diagrams can be quite complex. I don't think this would be an issue if the text contained more clear exposition of the method.

**Reviewer Expertise:**

Excellent: Expert knowledge on the topic of the paper

**Strengths And Weaknesses:**

Strengths
---
* Novel approach for sim2real/few-shot learning which bravely eschews end-to-end approaches
* Addresses difficult problems often avoided by this community, such as task-task adaptation and long-horizon tasks
* Compelling real and simulation experiments on challenging tasks
* Formulation appears very sample efficient, and extensions are easy to imagine (not a dead-end method)

Weaknesses
---
* Presentation issues make this difficult for readers to decipher, e.g. over-reliance on concepts for related works, prodigious use of bespoke acronyms and backronyms for many concepts (ASE, RBLD, SSR, CCD, PDP), complex diagrams demonstrations multiple concepts, tons of notation
* The paper struggles to help readers develop conceptual intuition for the methods before diving into immense amounts of detail

**Summary Of Recommendation:**

This paper presents an approach to the few-shot learning problem which diverges quite markedly from the current vogue, which centers on end-to-end and high-data techniques. End-to-end research tends to ignore long-horizon tasks and task-task adaptation, because it's very difficult to exploit the strengths of end-to-end learning to solve those (I have tried!).

After some laborious parsing, I believe the technique is technically sound and a novel, clever, and scalable approach to these neglected problems. I would love to see more research like this in the future, and I think original approaches like this one have a very high potential for impact.

---

### Official Review · Reviewer_tTpU · 2021-07-27

**Originality:** Good
**Technical Quality:** Good
**Clarity Of Presentation:** Fair
**Impact:** 3

**Recommendation:**

Weak Accept: I recommend accepting the paper, but will not argue for my recommendation if the majority of other reviewers have a different opinion.

**Summary:**


The paper proposes a novel algorithm for imitation learning using multi-task reinforcement learning pretraining. First, a set of policies is learned on the training tasks, together with Action-State Representations. The Action-State Representations are produced from a state using a neural network encoder, one for each action. At the imitation learning stage, the algorithm is always given the demonstration state-action corresponding to the current progress on the task. The algorithm finds the best action for the current state by maximizing the match between the Action-State Representation of the demo state-action pair and the current state-action pair. The match is evaluated using a cycle consistency loss. The performance of this method is slightly better than direct imitation learning.



**Issues:**

- How was the number of training steps for the Transporter tuned? What is the batch size? It appears that Transporter was trained on 12 data points for 40K iterations. Assuming a batch size of 12, that is 40K epochs. Is this training for this long really necessary?
- A lower bound baseline is missing. For instance, a very naive baseline would be executing random actions, or random actions taken from the demonstration trajectory. Another naive baseline would be repeatedly executing the demonstration trajectory until episode ends.

**Reviewer Expertise:**

Very good: Comprehensive knowledge of the area

**Strengths And Weaknesses:**


Strengths:
- The paper proposes a new system that is claimed to perform imitation learning, and empirically performs comparably to an imitation learning baseline (although using different assumptions).
- The paper proposes a new concept of state-action representations, which might be useful for future work.

Weaknesses:
1. The abstract contains numerous misleading statements.
1.1. For instance, it is claimed that SSR completes 20% more trials than the baseline. While technically a correct statement, this refers to a 6% improvement from 30% to 36%, which is the normal way of quantifying improvement and is perhaps less impressive.
1.2. Further, it is claimed that the method produces up to 3 orders of magnitude sample efficiency improvement. This is again correct: one of the baselines has the exact same data efficiency as the proposed method, but another baseline is 1000x worse. Another correct statement is that the proposed method produces between zero and 3 orders of magnitude sample efficiency improvement.
2. The proposed method only slightly outperforms the direct imitation learning baseline (Transporter). Given that the proposed method is significantly more complex, it is hard to justify this complexity. This baseline is also not included in the real world experiments. Since there is nothing special the proposed method does for real world experiments, the baseline should be also run the same exact way.
3. It is assumed that the policy observes its progress (i.e. how close to task completion it currently is) at test time. This is non-standard, and further a rather unrealistic assumption. Training the task completion estimator is technically as hard as training the policy to solve the task. Since the baseline methods don’t have access to task progress, the comparison is unfair.
4. The proposed imitation algorithm is simply presented mechanistically and it is entirely unclear what the design principles are.
5. The algorithm is specific to discrete actions
6. The system assumes the demonstrations are optimal (l151), which is often unrealistic.



**Summary Of Recommendation:**


The proposed method only slightly outperforms the direct imitation learning baseline (Transporter). Given that it is significantly more complex, it is hard to justify this complexity. The paper also suffers from many other issues with methodology, experiments, and writing, as outlined in my review. Overall, I am not convinced the proposed system represents a significant contribution to SOTA and leaning towards reject.

--- Update ---

In light of the otherwise positive updates to the paper (improved writing, improved performance, and the additional finetuned Transporter and SPOT-Q baselines), I switch my vote to weak accept. While the paper still suffers from certain writing issues, the experimental evaluation introduces extra assumptions, and the proposed method is rather cryptic, the updated results present a reasonable contribution for CORL.

---

### Meta-Review · Area_Chair_fi9M · 2021-08-03

**Recommendation:** Accept (Poster)
**Confidence:** 3

**Metareview:**

This paper presents an approach for learning from demonstrations that can transfer to new tasks, and includes experiments on challenging tasks, including on a real robot. Further, the related work section is thorough and the proposed ideas are interesting. These strengths are significant; however, the paper also has significant weaknesses:
* The proposed method assumes access to a task completion metric. This assumption is both non-standard and unrealistic, and is not justified in the paper. It is essentially akin to having access to a shaped reward function at test time. This assumption should be convincingly justified.
* The reviewers brought up multiple concerns about the fairness of the comparisons:
   * The TransporterNet comparison does not use the substantial pre-training data that is made available to the proposed method
   * None of the comparisons have access to the task completion metric, making it difficult to assess how much this assumption is helping the proposed method vs. other aspects of the proposed method
* Generally, the proposed method is fairly complex, and the improvement over TransporterNets is relatively small (30% success vs 36% success), making it unclear whether the complexity is worth it and whether future works will build upon this work in light of the complexity. It is strange that there is not a comparison to TransporterNets in the real world experiments.
* The clarity of the writing can be improved significantly to make it easier to follow and provide better intuition for the paper's design choices.

In the balance, while the paper has significant promise, the weaknesses of the paper currently outweigh the strengths.

-----------------------------------

-----------------------------------

After the response period, multiple of the concerns above have been addressed. There are still some important weaknesses, including (a) certain issues with writing / the description of the method and (b) being more upfront about assumptions like the task progress metric which does not come up until page 4 (or ideally experiments that learn this metric!). Further, I think the below statement in the abstract is not accurate:
> To our knowledge, we are the first to address multi-step tasks from demonstration on a real robot without task-specific training, where both the visual input and action space output are high dimensional

I believe that papers [1,2] also performed multi-step tasks from demos on a real robot without task specific training with visual inputs and high-dim actions, so this statement should be revised and these papers should potentially be referenced. I also believe that neither of these papers assume access to a task progress metric.

However, the strengths generally outweigh the weaknesses (despite these prior papers, this is still a very difficult problem setting!) and I recommend accept. Nonetheless, I strongly encourage the authors to make the changes listed above for the final version of the paper.


[1] Yu et al. IROS 2019 (https://arxiv.org/abs/1810.11043)

[2] Mandlekar et al. RSS 2020 (https://arxiv.org/abs/2003.06085)

---

### Decision · Program_Chairs · 2021-09-13

**Decision:**

Accept (Poster)

**Comment:**

This paper presents an approach for learning from demonstrations that can transfer to new tasks, and includes experiments on challenging tasks, including on a real robot. Further, the related work section is thorough and the proposed ideas are interesting. These strengths are significant; however, the paper also has significant weaknesses:
* The proposed method assumes access to a task completion metric. This assumption is both non-standard and unrealistic, and is not justified in the paper. It is essentially akin to having access to a shaped reward function at test time. This assumption should be convincingly justified.
* The reviewers brought up multiple concerns about the fairness of the comparisons:
   * The TransporterNet comparison does not use the substantial pre-training data that is made available to the proposed method
   * None of the comparisons have access to the task completion metric, making it difficult to assess how much this assumption is helping the proposed method vs. other aspects of the proposed method
* Generally, the proposed method is fairly complex, and the improvement over TransporterNets is relatively small (30% success vs 36% success), making it unclear whether the complexity is worth it and whether future works will build upon this work in light of the complexity. It is strange that there is not a comparison to TransporterNets in the real world experiments.
* The clarity of the writing can be improved significantly to make it easier to follow and provide better intuition for the paper's design choices.

In the balance, while the paper has significant promise, the weaknesses of the paper currently outweigh the strengths.

-----------------------------------

-----------------------------------

After the response period, multiple of the concerns above have been addressed. There are still some important weaknesses, including (a) certain issues with writing / the description of the method and (b) being more upfront about assumptions like the task progress metric which does not come up until page 4 (or ideally experiments that learn this metric!). Further, I think the below statement in the abstract is not accurate:
> To our knowledge, we are the first to address multi-step tasks from demonstration on a real robot without task-specific training, where both the visual input and action space output are high dimensional

I believe that papers [1,2] also performed multi-step tasks from demos on a real robot without task specific training with visual inputs and high-dim actions, so this statement should be revised and these papers should potentially be referenced. I also believe that neither of these papers assume access to a task progress metric.

However, the strengths generally outweigh the weaknesses (despite these prior papers, this is still a very difficult problem setting!) and I recommend accept. Nonetheless, I strongly encourage the authors to make the changes listed above for the final version of the paper.


[1] Yu et al. IROS 2019 (https://arxiv.org/abs/1810.11043)

[2] Mandlekar et al. RSS 2020 (https://arxiv.org/abs/2003.06085)